# Antitumor Activity of Simvastatin in Preclinical Models of Mantle Cell Lymphoma

**DOI:** 10.3390/cancers14225601

**Published:** 2022-11-15

**Authors:** Juliana Carvalho Santos, Núria Profitós-Pelejà, Marcelo Lima Ribeiro, Gaël Roué

**Affiliations:** 1Lymphoma Translational Group, Josep Carreras Leukemia Research Institute, 08916 Badalona, Spain; 2Laboratory of Immunopharmacology and Molecular Biology, Sao Francisco University Medical School, Braganca Paulista 01246-100, SP, Brazil

**Keywords:** statins, mantle cell lymphoma, cell proliferation, cell death

## Abstract

**Simple Summary:**

Mantle cell lymphoma (MCL) is an aggressive subtype of B-cell non-Hodgkin lymphoma that remains incurable with standard therapy. A promising approach to overcome the low life expectancy of patients with MCL is drug repurposing, a strategy focused on finding novel therapeutic weapons in approved drugs. Statins are well-tolerated, inexpensive, and widely prescribed as cholesterol-lowering agents that have also shown anti-cancer activity. The present study aimed to elucidate the effect of simvastatin on MCL cells. Our preclinical data demonstrate for the first time that simvastatin treatment impairs MCL proliferation and triggers a caspase-independent, ROS-mediated death of MCL cultures. Furthermore, we show that simvastatin significantly inhibits MCL migration and invasion ability, thereby impairing the growth of MCL tumors, suggesting that the use of statins might be considered for repurpose as a precise MCL therapy.

**Abstract:**

Background: Mantle cell lymphoma (MCL) is a rare and aggressive subtype of B-cell non-Hodgkin lymphoma that remains incurable with standard therapy. Statins are well-tolerated, inexpensive, and widely prescribed as cholesterol-lowering agents to treat hyperlipidemia and to prevent cardiovascular diseases through the blockage of the mevalonate metabolic pathway. These drugs have also shown promising anti-cancer activity through pleiotropic effects including the induction of lymphoma cell death. However, their potential use as anti-MCL agents has not been evaluated so far. Aim: The present study aimed to investigate the activity of simvastatin on MCL cells. Methods: We evaluated the cytotoxicity of simvastatin in MCL cell lines by CellTiter-Glo and lactate dehydrogenase (LDH) release assays. Cell proliferation and mitotic index were assessed by direct cell recounting and histone H3-pSer10 immunostaining. Apoptosis induction and reactive oxygen species (ROS) generation were evaluated by flow cytometry. Cell migration and invasion properties were determined by transwell assay. The antitumoral effect of simvastatin in vivo was evaluated in a chick embryo chorioallantoic membrane (CAM) MCL xenograft model. Results: We show that treatment with simvastatin induced a 2 to 6-fold LDH release, inhibited more than 50% of cell proliferation, and enhanced the caspase-independent ROS-mediated death of MCL cells. The effective impairment of MCL cell survival was accompanied by the inhibition of AKT and mTOR phosphorylation. Moreover, simvastatin strongly decreased MCL cell migration and invasion ability, leading to a 55% tumor growth inhibition and a consistent diminution of bone marrow and spleen metastasis in vivo. Conclusion: Altogether, these data provide the first preclinical insight into the effect of simvastatin against MCL cells, suggesting that this agent might be considered for repurpose as a precise MCL therapy.

## 1. Introduction

Mantle cell lymphoma (MCL) is a rare and aggressive subtype of B-cell non-Hodgkin lymphoma (B-NHL) that remains incurable with standard therapy and with a median overall survival of 5–7 years [1,2]. This malignancy accounts for about 5% to 8% of all lymphomas and can be clinically classified as conventional MCL with a more aggressive behavior, and non-nodal MCL, characterized by indolence with no or minimal lymphadenopathy and a low proliferation index [1]. The broad spectrum of clinical behaviors of this disease reflects the multistep alterations at the molecular level, starting from the driver event of t (11;14) (q13;q32) translocation leading to aberrant expression of cyclin D1, and cell cycle dysregulation, as well as individual genetic mutations with important therapeutic implications [3]. Although the initial treatment is not standardized, it usually includes non-curative chemotherapy regimens, depending on age and fitness of the patient. Nonetheless, relapsed and refractory (R/R) cases are frequent [2]. Thus, novel therapeutic options for MCL patients are urgently needed.

Simvastatin, lovastatin, pravastatin, and rosuvastatin belong to the statins class of drugs approved by the American Food and Drug Administration (FDA) to treat hyperlipidemia and to prevent cardiovascular diseases. These compounds block the mevalonate pathway, crucial for the synthesis of cholesterol, and as a result are one of the most widely prescribed cholesterol-lowering agents in the world. Statins lower plasma cholesterol levels by inhibiting 3-hydroxy-3-methylglutaryl coenzyme A (HMG-CoA) conversion to mevalonate and upregulating genes involved in cholesterol uptake [4,5]. Moreover, statins have been shown to inhibit prenylation of anti-apoptotic proteins and induce tumor specific apoptosis [6,7]. The pro-apoptotic effects of statins have been reported in a wide variety of cancers, such as breast, prostate, lung, glioblastoma, colorectal, osteosarcoma, acute myeloid leukemia, multiple myeloma, and lymphoma [8,9,10,11,12,13,14]. 

The growing effort to repurpose FDA-approved drugs to treat patients with cancer has prompted several studies to elucidate the pleiotropic anti-cancer effect of statins. Indeed, its use has been associated with low mortality and improved outcomes from several cancers, including hematological malignancies [15,16]. However, no clear evidence for their use in MCL has been shown to date. Remarkably, the effectiveness of statins is likely context-dependent and has been shown in specific subtypes of tumors or in combination with specific anti-cancer drugs. Here, we present a set of preclinical data and biological evidences obtained from in vitro and in vivo models of MCL, suggesting a potential use of simvastatin for the treatment of MCL patients. 

## 2. Materials and Methods

### 2.1. Simvastatin Activation

Simvastatin was activated by the opening of the lactone ring prior to use in cell assays. Briefly, a 10 mM stock solution of simvastatin (MedChemExpress, Monmouth Junction, NJ, USA) was dissolved in 100% ethanol, with the subsequent addition of 0.1 N NaOH. The solution was heated at 50 °C for 2 h and then neutralized to pH 7.2 in phosphate buffer saline (PBS). The aliquots were stored at −80 °C until use.

### 2.2. Cell Culture

Human MCL cell lines (REC-1, Z-138, JEKO-1, MINO, GRANTA-519 and UPN-1) were grown in Advanced-RMPI 1640 supplemented with 5% heat-inactivated fetal bovine serum (FBS), 2 mmol/L glutamine, and 50 µg/mL penicillin-streptomycin (Thermo Fisher Scientific, Waltham, MA, USA). JEKO-1, MINO, and REC-1 parental cells were obtained from ATCC cell bank (LGC Standards); GRANTA-519 cell line was purchased at DSMZ; and UPN-1 and Z-138 cells were provided by Dr. B. Sola (University of Caen, Caen, France). Cultures were routinely subjected to cell authentication and mycoplasma detection tests.

### 2.3. Cytotoxicity Assay

Cell viability and cytotoxicity were determined using CellTiter-Glo luminescent assay (Promega, Madison, WI, USA) and lactate dehydrogenase (LDH) release assay (Roche Diagnostics GmbH, Mannheim, Germany), respectively, according to the manufacturer’s instructions. Briefly, 3 × 10^4^ cells/well were cultured in sterile 96-well plates in the presence of increasing concentrations of simvastatin or vehicle in complete medium. After 24 or 48 h, the bioluminescence and absorbance were detected on a Synergy (BioTek, Winooski, VT, USA) microplate reader. 

### 2.4. Proliferation Rate Assays

The proliferation rate and mitotic indexes were determined by directly counting cell numbers over time, and by quantification of phospho-histone 3 (Ser10) positive cells, respectively. Briefly, REC-1 and Z-138 cells treated with simvastatin or vehicle were seeded on poly-L-lysine-coated glass coverslips, fixed with 4% paraformaldehyde in PBS pH 7.4 for 10 min at room temperature, and then permeabilized and blocked with PBS −0.3% Triton −5% FBS for 60 min. Cells were incubated with anti-phospho H3 (Ser10) antibody (Abcam, Cambridge, UK, #ab197502) at room temperature for 2 h, followed by an Alexa Fluor-conjugated goat anti-mouse IgG secondary antibody (Sigma, Merck, St. Louis, MO, USA) at room temperature for 1 h and the cover slips were mounted with anti-fading mounting medium containing 4,6-diamidino-2-phenylindole (DAPI). Fluorescence signal was acquired on a Leica microscope and quantified using the LAS X (Leica) and Image J softwares (ver.1.53K).

### 2.5. Apoptosis Rate and Mitochondrial Transmembrane Potential (ΔΨm)

REC-1 and Z-138 cell lines were treated with 5 μM, 10 μM or 20 μM simvastatin or vehicle for 48 h, and apoptosis was quantified by Annexin V-FITC/propidium iodide (PI, BD Biosciences, San Diego, CA, USA) staining. To assess if the apoptotic changes were caspase-dependent, cells were previously treated with 10 μM pan-caspase inhibitor, Q-VD-OPh hydrate (Sigma, Merck, St. Louis, MO, USA, SML0063). Briefly, cells were incubated with AnnexinV (Invitrogen, Thermo Fisher Scientific, Waltham, MA, USA) for 20 min at room temperature, washed and then incubated with propidium iodide (PI, BD Biosciences). Changes in mitochondrial transmembrane potential (ΔΨm) were assessed by staining cells with 20 nM 3,3′-diexyloxacarbocyanine iodide (DiOC6 [3]; Molecular Probes) for 15 min at 37 °C. In both analyses, 10,000 labeled cells per sample were acquired on a FACS CantoII flow cytometer (Becton Dickinson, San Jose, CA, USA) and analyzed using FlowJo software (Tree Star, Ashland, OR, USA).

### 2.6. Reactive Oxygen Species Quantification

A DHE (dihydroethidium) assay kit (MedChemExpress, Monmouth Junction, NJ, USA, HY-D0079) was used to measure ROS levels in live MCL cells cultured for 48 h in the presence or absence of 5 μM, 10 μM or 20 μM simvastatin. The percentage of red fluorescent (ROS-positive) cells was determined on the FACSCantoII flow cytometer using FlowJo software (Tree Star, Ashland, OR, USA).

### 2.7. Cell Migration and Invasion Assays

A 24-well Boyden chamber (8 μm pore size) transwell assay was used to investigate changes in cell motility. A 24-well Boyden chamber (8 μm pore size) covered with collagen I (Gibco, Thermo Fisher Scientific, Waltham, MA, USA) was used to assess cell invasion capacity. Briefly, REC-1 and Z-138 cells were grown on top of a transwell insert in serum-free media, in the presence or absence of 10 μM simvastatin. After 24 h, cells that migrated or invaded to the bottom of the lower chamber containing 600 μL of 10% FBS medium were quantified on the FACSCantoII flow cytometer.

### 2.8. Western Blot Analysis

Total protein extracts were collected from REC-1 and Z-138 cell lines treated with 5 μM, 10 μM or 20 μM simvastatin or vehicle. Then, 20 μg of protein was separated on 10% SDS-PAGE gel followed by transfer to a nitrocellulose membrane. After blocking with 5% milk powder in Tris-buffered saline (TBS)-Tween for 1 h, membrane-transferred proteins were revealed by incubating with primary and secondary antibodies followed by chemiluminescence detection using the ECL system (Pierce) and a Fusion FX imaging system (Vilber Lourmat, Marne La Vallee, France). The primary antibodies used were anti-AKT (Cell Signaling Technology, #9272), anti-phospho-AKT (Cell Signaling Technology, Danvers, MA, USA, #4060), anti-mTOR (Cell Signaling Technology, #2972), anti-phospho-mTOR (Cell Signaling Technology, Danvers, MA, USA, #2971), and β-actin (Santa Cruz, #sc-47778). The AKT and mTOR phosphorylation levels were evaluated by quantification of band intensity using Image J software and normalized to the corresponding total protein level. Values were referred to the indicated control and added below the corresponding band. 

### 2.9. Chick Chorioallantoic Membrane (CAM) Assay

Fertilized white Leghorn chicken eggs were purchased from Granja Santa Isabel, S. L. (Córdoba, Spain) and incubated for 9 days at 37 °C with 55% humidity. On day 9 of their embryonic development, a small window was drilled on top of the air chamber of the eggshell. Then, one million REC-1 cells were resuspended in 25 µL of RPMI complete medium (Thermo Fisher Scientific, Waltham, MA, USA) and 25 µL of Matrigel (BD Biosciences). After 15 min of incubation at 37 °C, the cells were inoculated into the CAM of each egg. In all, 10 μM simvastatin or vehicle diluted in RPMI medium was administered topically on the tumor-bearing CAMs on days 12 and 14 of chicken embryonic development. On the 16th day of development, chick embryos were euthanized by decapitation. Tumors were excised and carefully weighed to determine their mass. Then, tumor samples were formalin-fixed and paraffin-embedded prior to immunohistochemical staining with primary antibodies against CD20 (L26, Sigma) and phospho- Histone H3 (E173, Abcam, Cambridge, UK). Preparations were evaluated using an Olympus microscope and Micro-Manager Software (Tucson, AZ, USA).

### 2.10. Quantitative Real-Time PCR

DNA from chick embryo’s spleen and bone marrow was isolated according to manufacturer instructions using a Genomic DNA Purification Kit (Promega). The metastasis rate was evaluated by quantitative real-time PCR; the relative amount of human Alu sequences was quantified by the comparative cycle threshold method (ΔCt) using HPRT1 as an endogenous control. 

### 2.11. Statistical Analysis

Presented data are the mean ± SD or SEM of 3 independent experiments. All statistical analyses were done using GraphPad Prism 9.0 software (GraphPad Software). Comparison between two groups was carried out by unpaired t-test. Values of *p* < 0.05 were considered statistically significant. Results were considered statistically significant when *p*-value < 0.05.

## 3. Results and Discussion

### 3.1. Simvastatin Impairs the Proliferation and Triggers a Caspase-Independent ROS-Mediated Cell Death of MCL Cells

Growing evidence suggests that statins may be used as a potential cancer therapeutic strategy, depending on the type of tumor [17]. The anti-proliferative effect of statins on B-NHL has been evaluated in epidemiological studies with a remarkable lymphoma risk reduction [18,19]. To investigate their activity on MCL cell growth, we first examined the effect of different concentrations of simvastatin on the viability of a panel of 6 MCL cell lines (REC-1, Z-138, JEKO-1, MINO, GRANTA-519 and UPN-1). After a 48 h treatment, REC-1 and Z-138 appeared to be the two most sensitive MCL cell lines to simvastatin with respective IC50 of 4.97 µM and 3.78 µM, whereas the response of MINO, GRANTA-519, UPN-1 and JEKO-1 was less evident, as shown by IC50 values of 11.20 µM, 19.78 µM, 28.60 µM, and 39.81 µM, respectively (Figure 1A). Based on these results, REC-1 and Z-138 cell lines were selected for the subsequent experiments. The direct cytotoxicity of 5 μM, 10 μM and 20 μM simvastatin on MCL cells was then evidenced by a 3-fold increase in LDH release by the REC-1 cell line and a 2.3, 5.0 and 6.7-fold increase in LDH release by the Z-138 cell line, respectively, when compared to control cells (Figure 1B). These results were in accordance with previous data obtained in Hodgkin’s lymphoma [7] and mice lymphoma cells [8]. 

The PI3K/AKT/mTOR signaling pathway is one of the most often deregulated pathways in human cancers and plays a pivotal role in regulation of cell survival, growth, apoptosis, migration and metabolism [20]. Interestingly, simvastatin has previously been reported to alter lipid raft composition and induce apoptosis by decreasing AKT signaling in prostate cancer cells [21]. In support of these findings, we found that simvastatin treatment led to a 65–83% and 24–55% reduction in AKT and mTOR phosphorylation levels, respectively, in the REC-1 cell line, and to a 68–90% and 4–29% reduction in AKT and mTOR phosphorylation levels, respectively, in the Z-138 cell line, following a dose-dependent pattern (Figure 1C). This prompted us to investigate whether the antitumor activity of simvastatin could be ascribed to its ability to trigger a proliferation blockade and/or to elicit programmed cell death in our model. For this purpose, we carried out a direct cell counting and a mitotic index determination by phospho-H3 immunofluorescence of simvastatin-treated cell cultures. Our data revealed that the 5 μM, 10 μM and 20 μM doses of the drug were able to elicit a 0.31-, 0.16- and 0.08-fold reduction in REC-1 proliferation, as well as a 0.47-, 0.21- and 0.05-fold decrease in Z-138 cell growth, respectively (Figure 1D and Appendix A). This remarkable cell proliferation impairment was associated with a 4-fold and a 2 to 8-fold mitotic index blockade in REC-1 and in Z-138 cell lines, respectively (Figure 1E). These data were in accordance with the above-described disruption of AKT signaling and with previous studies demonstrating that the depletion of intermediates of mevalonate pathway, including farnesyl pyrophosphate (FPP) and geranylgeranyl pyrophosphate (GGPP), might influence the expression of genes involved in the regulation of cell proliferation and cell cycle progression [22,23].

We further evaluated the ability of 5 μM, 10 μM and 20 μM of simvastatin to trigger mitochondrial depolarization (ΔΨm loss) and phosphatidylserine ecto-exposure, by staining the cells with DiOC6 and AnnexinV-FITC/PI, respectively, followed by FACS analysis. As shown in Figure 1F and Appendix A, the number of cells with mitochondrial depolarization increased from 6.6% in REC-1 control cells, to 16.36%, 25.6% and 32.7% in REC-1 cells treated with 5 μM, 10 μM and 20 μM simvastatin, respectively. Similarly, mitochondrial depolarization was observed in 5.29% of untreated Z-138 cells, whereas this phenomenon affects 13.03%, 24.58% and 32.1% of Z-138 cells exposed to 5 μM, 10 μM and 20 μM simvastatin, respectively, suggesting a remarkable mitochondrial dysfunction in MCL cells induced by simvastatin. Interestingly, while Annexin V labeling revealed almost the same degree of activity for simvastatin in Z-138 and REC-1, the pan-caspase inhibitor Q-VD-OPh failed to rescue the ecto-exposure of phosphatidylserine evoked by the drug treatment (Figure 1G and Appendix A), suggesting that the cell death process elicited by simvastatin treatment was mainly caspase-independent. In this sense, we further demonstrated that simvastatin treatment led to the production of ROS, a phenomenon associated with different subtypes of caspase-independent processes in tumor cells; this was shown by a significant increase in DHE + cell fraction from 7.84% in the REC-1 control cells, to 27.96% and 28.86% in the cultures exposed to 10 μM and 20 μM simvastatin, respectively, and from 10.17% in Z-138 control cells to 27.7% and 38.2% after treatment with 10 μM and 20 μM simvastatin. On the other hand, no significant difference was observed in cells treated with the lowest (5 μM) dose of simvastantin (Figure 1H and Appendix A), suggesting that the apoptogenic properties of the drug at doses > 10 μM might involve the triggering of intracellular oxidative stress. Supporting these data, previous studies have shown that simvastatin inhibits proliferation of breast cancer cells along with an increase in ROS production, and a high dose of atorvastatin induces oxidative DNA damage in human peripheral blood lymphocytes [24,25]. Additionally, the induction of apoptosis by statins towards lymphoma cells has been shown to be associated with increased intracellular ROS generation, and suppression of activation of Akt, Erk and p38 signaling pathways during the suppression of mevalonate metabolic reaction [8]. 

### 3.2. Simvastatin Inhibits MCL Migration, Invasion and In Vivo Tumorigenic Properties

Statins treatment has been associated with impaired migration and invasion of breast cancer [26], pancreatic cancer [27], glioma cells [28], melanoma cells [29], renal cancer [30], prostate cancer [31], non-small cell lung cancer [32], and chronic lymphocytic leukemia (CLL) [33]. We used transwell assay to address whether simvastatin could also regulate MCL migration and invasion properties in vitro, using a 10 μM drug concentration based on the above results. Consistent with the previous observation found in other cancer types, our data show that simvastatin markedly inhibited 40–80% of MCL cell migration and 40–85% invasion capacity, respectively, in Z-138 and REC-1 cells (Figure 2A,B). In line with these results, increased membrane cholesterol levels in tumoral cells has been associated with the migration of several types of cancer, and therefore, simvastatin could regulate cell migration through lipid rafts modulation [34,35]. Similarly, a previous study showed that simvastatin can induce a decrease in CLL adhesion and invasion triggered by CXCL12 and CXCL13 chemokines, possibly by targeting HMGCR and LFA-1-mediated cell adhesion [33].

To further support the anti-lymphoma action of simvastatin in an in vivo setting, we established a chick embryo chorioallantoic membrane (CAM) xenograft model of MCL using the REC-1 cell line (Figure 2C). The CAM model has been used in cancer research for more than a hundred years to assess angiogenesis, metastasis, tumor growth, and drug responses [36,37,38]. This system has shown several advantages over other classical in vivo models, such as cost, time-effectiveness and simplicity. Importantly, chickens appear to have a highly coding and non-coding sequence conservation and an immune system functionally similar to humans [39]. After engraftment of MCL cells, the tumors were exposed twice to 10 μM simvastatin or an equivalent volume of vehicle. On day 16 post-egg fertilization, the chick embryos were sacrificed, and tumors were weighed. As shown in Figure 2D,E, the exposure to simvastatin led to a > 2-fold tumor weight decrease when compared to control tumors, along with a marked decrease of mitotic index, as revealed by reduced phospho-histone H3 levels in CD20 + tumor cells (Figure 2F). Moreover, simvastatin treatment strongly reduced the metastasis potential of MCL cells, evidenced by a > 2x drop in the abundance of human Alu sequences in the bone marrow and spleen of chick embryos (Figure 2G), further supporting the therapeutic activity of simvastatin against MCL in vivo.

## 4. Conclusions

In conclusion, our preclinical data demonstrate that the treatment with simvastatin impairs MCL cell proliferation, triggering ROS-mediated, caspase-independent cell death, associated with the suppression of the activation of the AKT/mTOR signaling pathway. Furthermore, simvastatin significantly inhibited MCL migration and invasion ability, thereby impairing the growth of MCL tumors, and the infiltration of chick embryo´s bone marrow and spleen by tumor B cells. Although additional in vitro and in vivo studies are warranted, our data suggest that the use of statins might be considered as a repurposing strategy for the design of a new precise MCL therapy.

## Figures and Tables

**Figure 1 cancers-14-05601-f001:**
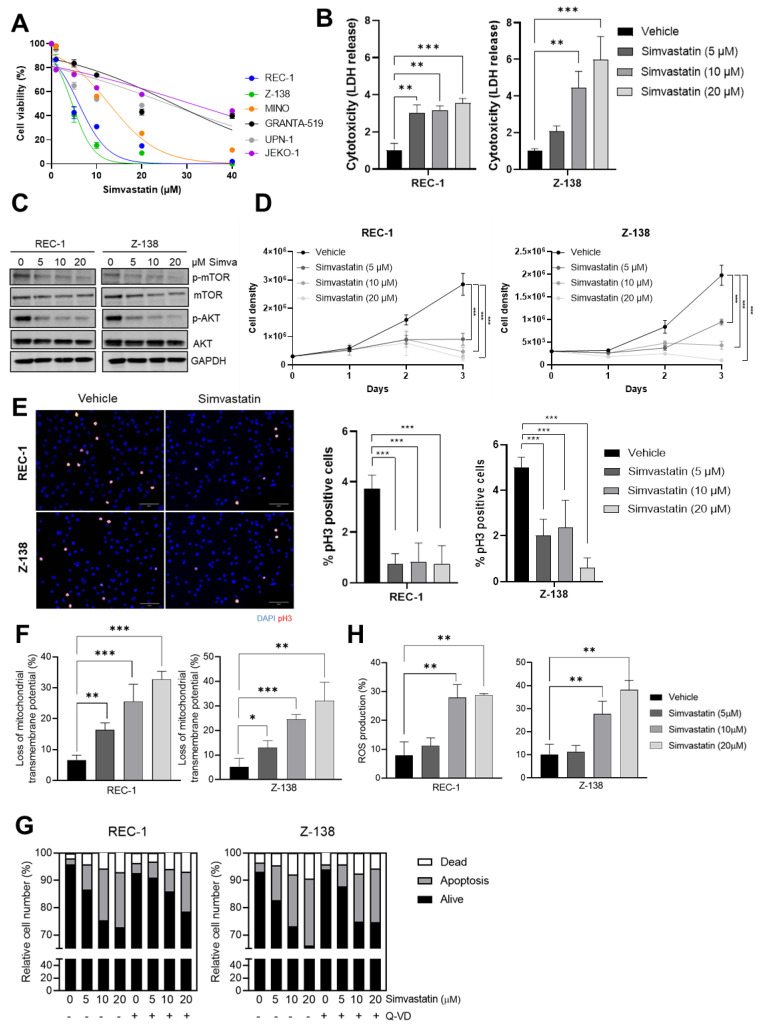
MCL cytotoxicity, proliferation impairment and cell death induced by simvastatin. (**A**) Cell viability of MCL cell lines in the presence or absence of different doses of simvastatin; (**B**) cytotoxicity rates assessed by LDH release of MCL cell lines in presence or absence of 5 μM, 10 μM and 20 μM simvastatin; (**C**) Western blot evaluation of phospho-AKT and phospho-mTOR levels after the treatment with 5 μM, 10 μM and 20 μM simvastatin, or vehicle. Full western blot images can be seen in Appendix A. (**D**) Proliferation rate assessed by MCL cell counting after the treatment with increasing doses of simvastatin; (**E**) proliferation rate assessed by phospho-Histone 3 (red) immunofluorescence of MCL cell lines in the presence or absence of 5 μM, 10 μM or 20 μM simvastatin. Nuclei were counterstained with DAPI (blue); (**F**) mitochondrial transmembrane potential (ΔΨm) after cell treatment with 5 μM, 10 μM or 20 μM simvastatin or vehicle; (**G**) apoptosis rate measured by AnnexinV-FITC/PI staining after cell treatment with 5 μM, 10 μM or 20 μM simvastatin or vehicle, previously treated or not with 10 μM of the pan-caspase inhibitor, Q-VD-OPh hydrate; (**H**) ROS generation measured by DHE (dihydroethidium) staining in MCL cells lines treated with 5 μM, 10 μM or 20 μM simvastatin, or vehicle. Values are expressed as mean ± SEM. * *p* < 0.05, ** *p* < 0.01, *** *p* < 0.001, when compared to control group.

**Figure 2 cancers-14-05601-f002:**
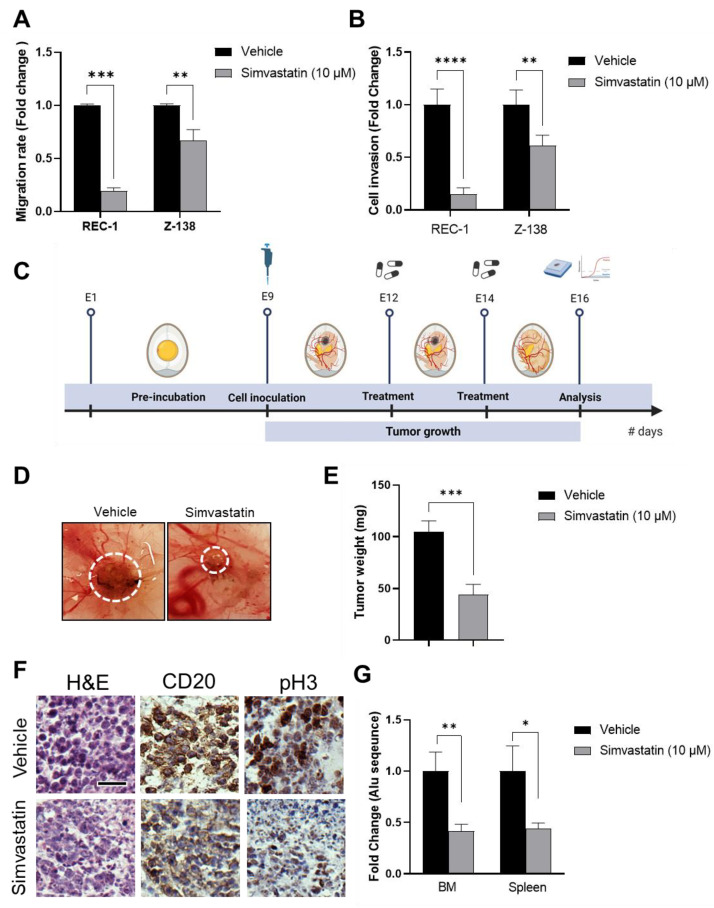
MCL migration and invasion ability, tumor growth and metastasis inhibited by simvastatin treatment. (**A**) Cell migration index of MCL cell lines exposed to 10 μM of simvastatin or vehicle for 24 h; (**B**) cell invasion index of MCL cell lines exposed to 10 μM simvastatin or vehicle for 24 h; (**C**) scheme depicting the chick embryo chorioallantoic membrane (CAM) model; (**D**) representative pictures of engrafted MCL tumors treated with 10 μM simvastatin or vehicle on day 16 of embryonic development. The dotted line delimitates the tumor; (**E**) tumor weight on day 16 of embryonic development after the treatment with 10 μM simvastatin or vehicle; (**F**) H&E and immunohistochemical (IHC) detection of CD20 and H3-pSer10 in tissue sections from CAM-tumor specimens dosed with 10 μM simvastatin or vehicle; (**G**) metastasis evaluation measured by qPCR-based Alu-sequence presence into embryo’s bone marrow and spleen. Values are expressed as mean ± SEM. * *p* < 0.05, ** *p* < 0.01, *** *p* < 0.001, when compared to the control group.

## Data Availability

Data presented in this study is maintained within this article.

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
