# Peer review of "Antitumor Activity of Simvastatin in Preclinical Models of Mantle Cell Lymphoma"

_cancers, 2022, doi:10.3390/cancers14225601_

Round 1

Reviewer 1 Report

1.       For Figure 1D, for the proliferation assay, author should add images with graph.

2.       For Figure 1C, author showed REC-1 cells line proteins after treatment, but in the other figures, they have Z-138. Why?

3.       Figure 1E, which cell line it is? Why only 1 cell line IF representation?

4.       For Figure1F, it’s good to have Flow data for mitochondrial membrane potential assay along with bar graph.

5.       Figure 2D, why did author use only one cell line (REC-1) for in vivo study?

6.       In the material and method, author should add qPCR method separately,

7.       Figure 1A, whys the points are way far from the line?

Author Response

Dear,

Thanks for giving us the opportunity to resubmit a substantial revised version of our manuscript. In our revised manuscript we have fully addressed your concerns and we are providing new exciting data to further support the treatment with simvastatin induces MCL caspase-independent ROS-mediated cell death, and proliferation impairment (Please see the attachment.)

We look forward to hearing from you.

Best regards,

Juliana Carvalho Santos, PhD

Reviewer 2 Report

The manuscript by Santos J.C. et. al. assessed the in vitro and in vivo anti-tumor efficacy of simvastatin in preclinical models of mantle cell lymphoma (MCL). Given the facts that statins are well-tolerated, inexpensive and widely prescribed as cholesterol-lowering agents to treat hyperlipidemia and to prevent cardiovascular diseases, and that statins showed anti-lymphoma activity, the authors hypothesized that simvastatin has the potential to treat MCL. The authors identified Rec-1 and Z138 cells (IC50 < ~ 5 uM) are two most sensitive cell lines among 4 others (IC50 >11 uM) and therefore these two cell lines were selected for further investigation. The authors found that simvastatin inhibits cell proliferation by inducing ROS-mediated cell death in caspase-independent manner. Furthermore, the authors showed evidence that simvastatin inhibits tumor growth and metastasis in vitro and in vivo.

Major points:

1.       Is there any ethics approval for the CAM assay? Please specify.

2.       Only a small fraction of the MCL cell lines (2 out of 6) showed sensitivity (IC50 < ~ 5 uM) to simvastatin. This may suggest that most of the MCL cases may not respond to simvastatin. For precision medicine, how to select patient for treatment if it does work in patients.

3.       Only one dose of simvastatin (10uM) was used in all in vitro experiments except Figure 1A and only one cell line was used for the western data in Figure 1C. Suggest to use at least two doses and two cell lines for the data in Figure 1.

4.       In the supplementary Figure, 10 uM simvastatin treatment resulted in 15.2% of Rec-1 cells and 19.5% of Z138 cells at early apoptosis (annexin V-positive), which are at higher percentages than those PI-positive, indicating that simvastatin can induced cell apoptosis. Simvastatin at 10 uM  reduced the annexin V-positive population in both cell lines. Simvastatin is typically used at higher concentration to inhibit cell apoptosis. Therefore, it is possible that 10uM simvastatin as a low dose is not sufficient to fully inhibit cell apoptosis. This experiment needs to be repeated with different doses of simvastatin together with other apoptosis inhibitors such as Z-VAD-FMK.

Minor points:

1.       Figure 1 legend: two panel Es. Legend for E-H appears to be mispresented.

2.       Typo: CellTiter-Glo, not CellTitter-Glo

Author Response

(The authors gave the same response as above.)

Round 2

Reviewer 2 Report

The authors have addressed all my concerns. Thanks.